# Antibiotic Resistance in Patients with Peri-Implantitis: A Systematic Scoping Review

**DOI:** 10.3390/ijerph192315609

**Published:** 2022-11-24

**Authors:** Carlos M. Ardila, Annie Marcela Vivares-Builes

**Affiliations:** 1Basic Studies Department, School of Dentistry, Universidad de Antioquia UdeA, Medellín 050010, Colombia; 2School of Dentistry, Institución Universitaria Visión de Las Américas, Medellín 050031, Colombia

**Keywords:** antibiotic resistance, microorganisms, peri-implantitis, systematic review

## Abstract

The implementation of adjunctive antibiotics has been recommended for the therapy of peri-implantitis (PI). In this review, antibiotic resistance patterns in PI patients were assessed. A systematic scoping review of observational studies and trials was established in conjunction with the PRISMA extension for scoping reviews. The SCOPUS, PubMed/MEDLINE, EMBASE, SCIELO, Web of Science, and LILACS databases were reviewed along with the gray literature. The primary electronic examination produced 139 investigations. Finally, four observational studies met the selection criteria. These studies evaluated 214 implants in 168 patients. *Porphyromonas gingivalis* and *Fusobacterium nucleatum* mainly presented high resistance to tetracycline, metronidazole, and erythromycin in PI patients. Similarly, *Aggregatibacter actinomycetemcomitans* was also highly resistant to clindamycin and doxycycline. Other microorganisms such as *Tannerella forsythia*, *Parvimonas micra*, and *Prevotella intermedia/nigrescens* also presented significant levels of resistance to other antibiotics including amoxicillin, azithromycin, and moxifloxacin. However, most microorganisms did not show resistance to the combination amoxicillin metronidazole. Although the management of adjunctive antimicrobials in the therapy of PI is controversial, in this review, the resistance of relevant microorganisms to antibiotics used to treat PI, and usually prescribed in dentistry, was observed. Clinicians should consider the antibiotic resistance demonstrated in the treatment of PI patients and its public health consequences.

## 1. Introduction

Peri-implantitis (PI) is a destructive pathological complication appearing in soft and hard tissues surrounding dental implants, which is expressed by the inflammation of the peri-implant mucosa and the consequent gradual loss of alveolar bone [1]. The prevalence of PI ranges from 22% to 45% [2], and PI is predicted to appear in 11–47% of dental implant subjects 10 years after prosthodontics [3].

In the PI development process, bacterial occupation of the implant surface is the principal causal agent [4]. The biofilm of PI comprises a complicated collection of microorganisms that generate the infection and induce the associated illness [5]. The microbiota in PI appears to be predominantly constituted by anaerobic Gram-negative bacteria and is not related to a constant bacterial outline, which is different to periodontal diseases [6]. A manifest dissimilarity has not been distinguished in the microbial species concerning periodontitis and PI [7]. Nevertheless, various microorganisms, such as *Tannerella forsythia*, *Porphyromonas gingivalis*, and *Prevotella intermedia*, among others, have been observed in vast quantities in PI [7]. This group of microorganisms has been correlated with the beginning of PI [8,9]. In peri-implant health, these species are in balance with the host; nonetheless, the occurrence of several risk factors that aid biofilm creation might generate the modification of the microenvironment and tissue swelling. Therefore, this situation, in combination with deficient oral hygiene, can raise the amounts of these bacteria [10]. The growth of complicated infectious microbiota exemplifies clinical defiance in PI treatment [11]. The irregular progressive configuration of bone loss in PI causes implant failure, wherein the known infection is not competently contained [12]. Thus, the management of PI supports the control of infection and decreases microbial burden.

Since the etiopathogenesis of PI is not properly explained, its most efficient treatment has yet to be categorically recognized [13]. However, several PI management protocols have been described, including mechanical therapy, surgery, and mixed methods. Mechanical management alone has presented uncertain treatment results, whereas effects following surgical therapy present just a slight increase in bone level [14,15]. Notably, several systematic reviews have indicated that most surgical PI procedures recommend the adjunctive implementation of systemic antimicrobials to target putative microorganisms [15,16,17,18]. Moreover, a recent systematic review found that in healthful subjects’ antimicrobial prophylaxis is prescribed to avoid early implant failure [19]. It has also been reported that most clinicians empirically use antimicrobials against PI without a bacteriological examination [20,21]. Different antimicrobials have also been proposed in dentistry for surgical and non-surgical use with local administration [22,23]. The drawbacks mainly include enhancing antimicrobial resistance and the risk of altering the normal microflora, as well hypersensitivity reactions, among others [24]. Moreover, various investigations have reported that adjunctive antimicrobials in regular dental procedures, principally combination management, can substantially contribute to the advancement of antibiotic resistance [25]. The elimination of a habitual, sensitive microbiota can also lead to greater resistance and additional infective microorganisms in the oral cavity [26]. Due to the risk of bacteriological resistance and the consequences for the whole human microbiome, which is intricately related to systemic antimicrobial treatment, the empirical implementation of these medicines must be completely discussed, specifically due to the increased benefit and plausible favorable outcomes that will likely result for the patient. Considering that merely specific dental patients have a pertinent and clinically relevant advantage after the intake of adjunctive antimicrobials, these prescriptions must be judiciously and restrictively applied [27]. Further, the global attention towards reducing the administration of antibiotics has led to novel pathways for antibiotic resistance [28,29].

Bearing in mind that the empirical line for the management and practice of antimicrobials in odontology is an established concern [30], it is appropriate to complete a systematic scoping review that facilitates the assessment of the antibiotic resistance patterns in PI patients.

## 2. Materials and Methods

### 2.1. Data Origin and Exploration Scheme

A systematic scoping review of observational and experimental studies was undertaken considering the PRISMA (Preferred Reporting Items for Systematic Reviews and Meta-analyses) addition for scoping reviews [31]. SCOPUS, PubMed/MEDLINE, EMBASE, SCIELO, Web of Science, and LILACS databases were reviewed along with the gray literature. MeSH terms and keywords were utilized to examine clinical investigations in all languages published until October 2022, containing the expressions peri-implantitis, dental implants, antibiotic resistance, antimicrobial resistance, bacterial resistance, antibiotics, antibiotics susceptibility, adjunctive antibiotics, randomized clinical trials (RCTs), and observational clinical investigations. Later, an examination mechanism was applied to review databases using Boolean operators (AND; OR): “peri-implantitis” OR “dental implants” AND “antibiotic resistance” OR “antimicrobial resistance” OR “bacterial resistance” OR “antibiotic susceptibility” AND “adjunctive antibiotics” AND “clinical research” OR “observational clinical studies” OR “RCTs”.

### 2.2. Selection Criteria

Inclusion criteria: Systematically healthy patients diagnosed with PI and managed with systemic or local antibiotics involving phenotypic exploration and antimicrobial sensitivity outcomes were included. Moreover, only observational clinical studies and clinical experiments were evaluated.

Exclusion criteria: Studies that observed nursing and pregnant females, involved subjects with systemic conditions, and involved the intake of antimicrobials within the previous 30 days were excluded. Case reports, case series, investigations in animals, and duplicate reports were also excluded.

### 2.3. Research Questions

The objective of this review is to answer to the following questions. In subjects displaying PI, (a) what is the prevalence of antibiotic-resistant species, and (b) to which antibiotics did the microorganisms studied show resistance?

### 2.4. Records Choice

Designated terms used by the two researchers facilitated the selection of publications founded on the evaluation of summaries and full texts. Successively, both investigators chose the studies after agreeing upon the formerly recognized reasons behind the selection. Then, all abstracts and full texts were independently evaluated. The suitability forms were applied to organize the articles to be integrated into this study. If disagreement between the researchers was observed, report’s suitability was resolved by consensus. The Kappa check was employed to express an agreed result among investigators (>90).

### 2.5. Information Extraction

The papers that matched the selection standards were evaluated independently by the two investigators and evaluated to organize this review. A chart was made that integrated the most relevant data from the chosen reports (separately by each of the authors), and the outcomes were compared. The table contains the authors’ information; year of publication; the quantity of subjects/implants; principal features of the methodology; the occurrence of antimicrobial-resistant species; and the antibiotic to which resistance was detected.

### 2.6. Outcome Measures

The main result was the frequency of antibiotic-resistant bacteria. The secondary result was the antibiotic to which the microorganism showed resistance.

### 2.7. Risk of Bias

The researchers independently evaluated the scientific grade of the selected reports utilizing a formerly validated tool that encompasses 16 criteria [32]. The investigators had to provide a result corresponding to a range from 0 to 3 for each of the conditions. If investigators did not describe in detail the aspects required to explain the reasoning behind a given criterion, an outcome of 0 was given to that point. If the feature was adequately described, a score of 3 was appointed. If outcomes were imprecise, a score of 2 was applied. By adding all the information, the proportion of the maximum possible score was obtained (100%).

## 3. Results

The primary examination produced 139 investigations, of which 111 were omitted since they did not meet the selection criteria. After examining the titles and summaries, another 12 reports were discarded. After detailing the full text of the publications, another 12 reports were removed. Finally, four cross-sectional studies [33,34,35,36] were analyzed in this study (Figure 1). No experimental study evaluated antibiotic resistance in PI patients.

The features of the included investigations are depicted in Table 1. These reports were available among 2014 [33] and 1995 [36]. The reports evaluated 214 implants in 168 subjects, with sample sizes from 11 [34] to 120 [33] individuals.

Subgingival plaque samples were taken in all studies. As seen in Table 1, most of the investigations used different culture methodologies to identify the microorganisms studied. Two studies used the same methodology to evaluate antimicrobial resistance [17,18,33,34], while one investigation used the E-test [35] and another implemented the minimum inhibitory concentration [36] based on previously established standards [37].

Table 1 depicts the prevalence of antibiotic-resistant microorganisms. *P. gingivalis* and *Fusobacterium nucleatum* showed great resistance to tetracycline, metronidazole, and erythromycin [36], and to clindamycin to a lesser extent [33,34]. *P. intermedia* was also very resistant to erythromycin [36]. Similarly, *Aggregatibacter actinomycetemcomitans* was highly resistant to clindamycin and doxycycline [33]. Other microorganisms also presented significant resistance values. It is pertinent to underline that a report described that 72% of the 120 PI patients included in the study displayed submucosal microorganisms resistant to one or more of the examined antibiotics [33].

Different species showed resistance mainly to adjunctive metronidazole [33,36] or doxycycline [33,34] in two studies, while resistance to amoxicillin [33], clindamycin [33], tetracycline [36], erythromycin [36], moxifloxacin [35], azithromycin [35], penicillin [35], and ampicillin [35] was reported in one study for each antibiotic (Table 1). However, most of the studies reported the susceptibility of the microorganisms studied to the combination of amoxicillin and metronidazole.

All studies assessed in this review fulfilled the majority (75%) of the defined criteria [32]; accordingly, they had good quality (Table 2). Nevertheless, these reports have substantial heterogeneity, which is principally observed in their designs, in the investigation of diverse types of drugs, and differences in the microbiological identification and the microorganisms reviewed, which makes it challenging to complete a suitable statistical analysis.

## 4. Discussion

This review evaluated the prevalence of antimicrobial-resistant species in PI patients. Although the management of adjunctive antimicrobials in the therapy of patients with PI is controversial, some systematic reviews have evaluated different clinical trials carried out recently [17,24]. This means that clinicians currently use antibiotics for the treatment of PI. Moreover, another recent systematic review also reported the evaluation of a relevant quantity of experiments that recommend the utilization of antibiotic prophylaxis during surgical procedures performed to install dental implants [19]. This corroborates the current use of antibiotics in the practice of implantology. Therefore, concern about antibiotic resistance is also present.

Herein, *P. gingivalis*, *A. actinomycetemcomitans*, *F. nucleatum*, and *P. intermedia/nigrescens* displayed high resistance to tetracycline, doxycycline, metronidazole, clindamycin, and erythromycin [33,34,36]. A recent systematic review corroborates these results in patients with periodontitis [38].

Resistance to tetracycline and tetracycline products may be transferred through “efflux pumps, ribosomal protection proteins” and inactivation enzymes. Tetracycline resistance genes code for these types of interaction and are usually present in mouth bacteria [23,39]. It has also been described that the abnormal magnitude of points showing *A. actinomycetemcomitans* and *P. gingivalis’s* resistance to tetracycline was correlated with a grander consumption of antimicrobials by the investigated community [40].

As noted in this review, an elevated amount of Gram-negative, anaerobic-resistant samples has been identified against metronidazole. This finding might be the effect of metronidazole’s restricted range of action [41].

Low susceptibility to erythromycin was also previously described in endodontic infections [42], and the augmentation of anaerobic resistance to clindamycin was also indicated in a recent systematic scoping review [42] with outcomes consistent with the great resistance of *Prevotella* spp. against penicillin [43]. These findings are predominantly caused by the facility for beta-lactamase production among anaerobes [42].

Moreover, it has also been described that the resistance to tetracycline, azithromycin, and metronidazole was comparable prior to and following therapy [41]. The in vitro reports presented equivalent effects at baseline [44]. Troublingly, it was described that *F. nucleatum* tended to extend its resistance to several antibiotics over time [43]. In this context, it has been affirmed that anaerobic sensitivity to antimicrobials can fluctuate due to geographic factors involving the prescription schedule, the antibiotic chosen, the breakpoint considered, and patient fulfillment [30,38,44].

Antibiotic resistance is an international challenge, wherein several nations are demonstrating greater resistance [42,44]. With this background, it is pertinent to emphasize that a current investigation denoted that antimicrobial-resistance genes were perceived in the plaque of more than 50% of healthy persons [45]. This aspect can partially elucidate the decreasing efficacy of the antimicrobials implemented clinically [46].

An additional issue that should be considered is the occurrence of cross-adaptation to antiseptics and antibiotics [47]. Thus, the augmented utilization of antiseptics has increased its risk, raising alarms concerning the consequential cross-resistance to additional antibiotics [48]. Troublingly, a recent investigation also explained that the quantity of subjects harboring partially resistant microorganisms expanded by 33% over time, and 63% of them presented non-sensitivity to at least one of the antimicrobials explored [49]. This shows that dentists must continuously assess the resistance of these microorganisms.

As noted in this review, the low incidence of PI subjects’ microbial resistance to the amoxicillin–metronidazole combination was comparable to that recently observed among subgingival microorganisms in periodontal diseases [38] and is coherent with the results of the enhanced PI clinical parameters potentiating the systemic administration of these antibiotics [50]. It was also observed that this combination was highly active against PI samples of *P. gingivalis*, *P. intermedia*, *and F. nucleatum.* These outcomes suggest that the results of adjunctive amoxicillin plus metronidazole in PI therapy can mirror the mixed medication’s efficiency in improving periodontal diseases treatment [51].

This systematic scoping review has some limitations. First, the antibiotic sensitivity outcomes in vitro do not essentially confer clinical efficacy against the target microorganisms [52]; therefore, the magnitude to which the informed laboratory results can hypothetically influence the therapy of PI is unknown [33]. Nevertheless, it is commonly documented that antibiotics lacking effectiveness against targeted bacteria under in vitro settings are unlikely to be clinically successful [53,54]. On the other hand, methodological differences were observed in the selected investigations, and their design does not permit a causal correlation to be established. It is also reasonable that the antibiotic-resistant microorganisms that had developed could not be recognized by the protocols implemented. Similarly, the prescription and the time of application of the antibiotic could have been inadequate, or the dissemination of the drug to the specific point could have been less than ideal [41]. Furthermore, geographic discrepancies in antibiotic resistance outlines appear [38,44,49,55]. Therefore, it is essential to carry out clinical trials that allow for the evaluation of longitudinal changes in antibiotic resistance in patients with PI.

## 5. Conclusions

Considering the strengths and boundaries of this novel systematic scoping review, it was observed that while the use of adjunctive antimicrobials in the management of PI is debated, the resistance of the recognized microorganisms to the antibiotics implemented to treat PI, and frequently prescribed in dentistry, was perceived. The conclusions of the current study can help clinicians and leaders of public health associations to create relevant proclamations and gain insight into the relevance of the administration of antibiotics. Clinicians must reconsider the benefits of antimicrobial treatment in the management of PI and ponder the potential consequences of antimicrobial resistance in the population.

## Figures and Tables

**Figure 1 ijerph-19-15609-f001:**
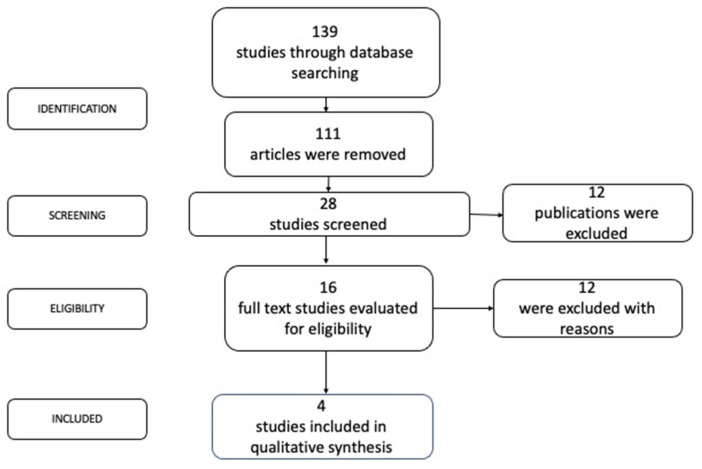
Search strategy for antibiotic resistance in PI.

**Table 1 ijerph-19-15609-t001:** Characteristics of the assessed studies.

Authors	Participants/Implants	Mean Age	Methodology	Prevalence of Antibiotic-Resistant Species	Antibiotic
Rams et al. 2014 [33]	120/160	61 years	Complete anaerobically viable amounts were determined on EBBA primary isolation plates, implementing reasonable phenotypic devices formerly defined. Resistance to the drug breakpoint intensities was documented when test bacteria growth was noticed on the antibiotic-supplemented EBBA plates.	Four Pg strains (3.3% of patients) presented resistance to clindamycin. Pi/n was resistant to amoxicillin among 38 strains (40% of patients), and clindamycin in 35 strains (47% of patients). Strains also showed resistance to doxycycline (25%) and metronidazole (22%). Six Tf strains showed resistance to amoxicillin (5% of patients) and clindamycin among seven strains (6% of patients). Sc strains had resistance to metronidazole (22%), 25% to doxycycline, 40% to amoxicillin, and 47% to clindamycin. All Aa strains presented resistance to clindamycin, and 83% to doxycycline.	Clindamycin Amoxicillin Metronidazole Doxycycline
Rams et al. 2013 [34]	11/11	74 years	Entire anaerobically viable amounts were recognized on EBBA plates utilizing probable phenotypic processes and standards formerly presented. Resistance to the drug breakpoint intensities was detailed when level of test bacteria growth was high on the antimicrobial-supplemented EBBA plates.	Fn exhibited resistance to doxycycline in one patient (9%).	Doxycycline
Karbach et al. 2013 [35]	24/24	NR	Microbial isolates were divided according to their cellular morphology. All isolates were Gram-stained. Aliquots of 0.1 mL were marked on agar plates to establish the probable resistance of the diverse samples. The sensitivity of the samples was confirmed utilizing the E-test.	Pm, Hs, and Ss were resistant to azithromycin, moxifloxacin, penicillin G, ampicillin, and ampicillin+sulbactam (28%). Two Aa (8%) and two As (8%) isolates presented resistance to all five tested antibiotics.	Azithromycin Moxifloxacin Penicillin G Ampicillin Ampicillin+Sulbactam
Sbordone et al. 1995 [36]	13/19	NR	Bacterial cultural examination of the species was achieved implementing continuous anaerobic procedures. The MICs of antimicrobials were established for predominant cultivable microflora. MICs were completed by the agar dilution system.	Pg (90%) presented resistance to tetracycline, metronidazole, and erythromycin. Pi was resistant to erythromycin (100%). Fn (90%) showed resistance to tetracycline, metronidazole, and erythromycin.	Tetracycline Metronidazole Erythromycin.

EBBA = enriched Brucella blood agar; Pg = Porphyromonas gingivalis; Tf = Tannerella forsythia; Pi/n = Prevotella intermedia/nigrescens; Fn = Fusobacterium nucleatum; Pm = Parvimonas micra; Hs = Haemophilus species; Ss = Streptococcus species; Aa = Aggregatibacter actinomycetemcomitans; As = Acinetobacter species; Sc = Streptococcus constellatus; MIC = minimum inhibitory concentration; NR = Not reported.

**Table 2 ijerph-19-15609-t002:** Feature of the selected reports [16].

Investigation	1	2	3	4	5	6	7	8	9	10	11	12	13	14	15	16	Result
Rams et al. [33]	3	3	3	0	0	3	3	3	3	3	3	3	3	3	0	3	81%
Rams et al. [34]	3	3	3	0	0	3	3	3	3	3	3	3	3	3	0	3	81%
Karbach et al. [35]	3	3	3	0	0	3	3	3	3	3	3	3	3	3	0	0	75%
Sbordone et al. [36]	3	3	3	0	0	3	3	3	3	3	3	3	3	3	0	3	81%

1—Clear academic background. 2—Declaration of purposes. 3—Study location. 4—Sample magnitude. 5—Representative sample. 6—Explanation of the method or data compilation. 7—The justification for the select of information compilation. 8—Comprehensive enrollment information. 9—Statistical valuation of consistency. 10—Coherence between specified study’s interrogation and methodology. 11—Consistency between indicated study question and content of records. 12—Coherence between specified research interrogation and process of analysis. 13—The investigative method designated. 14—Consistency of systematic method. 15—Operator participation in the project. 16—Strengths and boundaries.

## Data Availability

The data obtained in this review were pooled from the included investigations.

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
