# Peer review of "Antibiotic Resistance in Patients with Peri-Implantitis: A Systematic Scoping Review"

_ijerph, 2022, doi:10.3390/ijerph192315609_

Round 1

Reviewer 1 Report

In this work the authors assessed antibiotic resistance patterns in peri-implantitis patients.
The topic is interesting and should be encorauged.

-at line 30 the phrase should be modified from "..of post-therapy evaluation" into "..after prosthodontics"

-at line 38 the concept of different antibiotics administration should be more underlined: please add this phrase at line 38

".. Different antimicrobials have been proposed in dentistry for surgical and non-surgical use with local administration.."

please cite the following

Busa A, Parrini S, Chisci G, Pozzi T, Burgassi S, Capuano A. Local versus systemic antibiotics effectiveness: a comparative study of postoperative oral disability in lower third molar surgery. J Craniofac Surg. 2014;25(2):708-9. doi: 10.1097/SCS.0000000000000431. PMID: 24469372.

Graziani F, Karapetsa D, Alonso B, Herrera D. Nonsurgical and surgical treatment of periodontitis: how many options for one disease? Periodontol 2000. 2017 Oct;75(1):152-188. doi: 10.1111/prd.12201. PMID: 28758300.

-at line 43 the global intention in reducing antibiotics administration should be underlined, please add the following phrase:

"..Further, global attention to reduce antibiotics administration lead to different consideration for antibiotics resistance..."

please cite the following

D'Ambrosio F, Di Spirito F, De Caro F, Lanza A, Passarella D, Sbordone L. Adherence to Antibiotic Prescription of Dental Patients: The Other Side of the Antimicrobial Resistance. Healthcare (Basel). 2022 Aug 27;10(9):1636. doi: 10.3390/healthcare10091636. PMID: 36141247; PMCID: PMC9498878.

Chisci G, Hatia A. Antibiotics in orthognathic surgery and postoperative infections. Int J Oral Maxillofac Surg. 2022 Aug 27:S0901-5027(22)00320-4. doi: 10.1016/j.ijom.2022.08.008. Epub ahead of print. PMID: 36041952.

-figure 1 the description of the figure should be comprehensive of the image

Author Response

Response to the decision letter

Dear Reviewer 1: we are grateful for the constructive comments you provided, which helped us to improve the manuscript significantly.

Our responses to your comments are outlined below and highlighted in blue (to differentiate them from responses to other reviewers) in the new version.

Responses to Reviewer 1

  1. At line 30 the phrase should be modified from ". of post-therapy evaluation" into "..after prosthodontics"

Response:  The recommendation was included.

  1. At line 38 the concept of different antibiotics administration should be more underlined: please add this phrase at line 38. ".. Different antimicrobials have been proposed in dentistry for surgical and non-surgical use with local administration."

Response: The recommendation was included.

  1. Please cite the following: Busa et al. 2014; Graziani et al. 2017

Response: The recommendation was included.

  1. At line 43 the global intention in reducing antibiotics administration should be underlined, please add the following phrase: "..Further, global attention to reduce antibiotics administration lead to different consideration for antibiotics resistance..."

Response: The recommendation was included.

  1. Please cite the following: D'Ambrosio et al. 2022; Chisci et al. 2022

Response: The recommendation was included.

  1. Figure 1. The description of the figure should be comprehensive of the image.

Response: The recommendation was included.

Reviewer 2 Report

Dear authors,

The paper submitted by Ardilla et a. approaches a very interesting and contemporary topic.

I hope that my remarks will be useful in order to increase its’ quality.

1.     Usually „:” are not preffered in the titles.

2.     Abstract should be extended or at least re-organised in order to better emphasise the aim and the perspectives of this review.

3.     Figure 1 should be redesigned in a more academic manner.

4.     Table 1, although very interesting, is difficult to follow. The pieces of information should be more concise.

5.     I would suggest to include the following references in the discussion section in order to extend it:

Schmid, J.-L.; Kirchberg, M.; Sarembe, S.; Kiesow, A.; Sculean, A.; Mäder, K.; Buchholz, M.; Eick, S. In Vitro Evaluation ofAntimicrobial Activity of Minocycline Formulations for Topical Application in Periodontal Therapy. Pharmaceutics 2020, 12, 352.

Luchian, I.; Goriuc, A.;Martu, M.A.; Covasa, M. Clindamycin as an Alternative Option in Optimizing Periodontal Therapy. Antibiotics 2021, 10, 814.

Chambrone, L.; Vargas, M.; Arboleda, S.; Serna, M.; Guerrero, M.; de Sousa, J.; Lafaurie, G.I. Efficacy of local and systemic antimicrobials in the non-surgical treatment of smokers with chronic periodontitis: A systematic review. J. Periodontol. 2016, 87,1320–1332.

Jepsen, K.; Jepsen, S. Antibiotics/antimicrobials: Systemic and local administration in the therapy of mild to moderately advancedperiodontitis. Periodontology 2000 2016, 71, 82–112.

Herrera, D.; Matesanz, P.; Martín, C.; Oud, V.; Feres, M.; Teughels, W. Adjunctive effect of locally delivered antimicrobials in periodontitis therapy: A systematic review and meta-analysis. J. Clin. Periodontol. 2020, 47, 239–256.

6.     The Conclusion section should be rephrased and extended in such a way that will emphasise the perspectives of your review, its’ novelty and clinical potential impact.

Best regards!

Author Response

Response to the decision letter

Dear Reviewer 2: we are grateful for the constructive comments you provided, which helped us to improve the manuscript significantly.

Our responses to your comments are outlined below and highlighted in green (to differentiate them from responses to other reviewers) in the new version.

Responses to Reviewer 2

  1. Usually „:” are not preferred in the titles.

Response:  The recommendation was edited.

  1. Abstract should be extended or at least re-organized to better emphasize the aim and the perspectives of this review.

Response: The recommendation was included. The reviewer must consider that the journal only allows 200 words, which is the exact number that the abstract contains in the new version.

     3. Figure 1 should be redesigned in a more academic manner.

Response: The recommendation was amended. It is important to highlight that the figure follows the 4-phase flow recommended by the PRISMA statement: Moher, D., Liberati, A., Tetzlaff, J., Altman, D. G., & PRISMA Group (2009). Preferred reporting items for systematic reviews and meta-analyses: the PRISMA statement. Journal of Clinical Epidemiology, 62(10), 1006–1012.

  1. Table 1, although very interesting, is difficult to follow. The pieces of information should be more concise.

Response: In the revised version the information is more concise.

  1. I would suggest including the following references in the discussion section to extend it: Schmid et al. 2020; Luchian et al. 2021; Chambrone et al. 2016; Jepsen et al. 2016; Herrera et al. 2020.

Response: Some authors were considered. It is important to highlight that this review deals with antibiotic resistance in peri-implantitis. All recommended references are on antibiotic periodontal therapy, and none address antibiotic resistance. As the reviewer can see, most of the recommended authors are cited in other publications related to the topic discussed in this manuscript.

  1. The conclusion section should be rephrased and extended in such a way that will emphasize the perspectives of your review, its’ novelty and clinical potential impact.

Response: The recommendation was included.

Reviewer 3 Report

Dear authors,

Please put the references according to mesh terms and alphabetical orders.

Remove we and our from the manuscript.

A background should be done about the topic before the objective of the study.

More databases should be searched so you can extend your analyzed data

The introduction is too small and lacks a lot of informations about the topic.

Please change the flow chart into one figure

Discussion section is too weak

Add more limitations

Add more references

Author Response

Response to the decision letter

Dear Reviewer 3: we are grateful for the constructive comments you provided, which helped us to improve the manuscript significantly.

Our responses to your comments are outlined below and highlighted in yellow (to differentiate them from responses to other reviewers) in the new version.

Responses to Reviewer 3

  1. Please put the references according to MeSH terms and alphabetical orders.

Response:  The recommendation was included.

  1. Remove we and our from the manuscript.

Response: The recommendation was contemplated.

  1. A background should be done about the topic before the objective of the study.

Response: The recommendation was included.

  1. More databases should be searched so you can extend your analyzed data

Response: Additional databases were reviewed (EMBASE and Web of Science); however, the results obtained were replicated because the same studies met the selection criteria.

  1. The introduction is too small and lacks a lot of information about the topic.

Response: The introduction was amended.

  1. Please change the flow chart into one figure.

Response: The recommendation was included. It is important to highlight that the figure follows the 4-phase flow recommended by the PRISMA statement: Moher, D., Liberati, A., Tetzlaff, J., Altman, D. G., & PRISMA Group (2009). Preferred reporting items for systematic reviews and meta-analyses: the PRISMA statement. Journal of Clinical Epidemiology, 62(10), 1006–1012.

  1. Discussion section is too weak. Add more limitations. Add more references.

Response: Recommendations were followed. Now the manuscript contains 20 more references.

Reviewer 4 Report

The authors must be added a section concerning the future perception of peri-implantitis treatment and antibiotic resistance.

The authors must be added a section concerning the recommendation for peri-implantitis treatment

Author Response

Response to the decision letter

Dear Reviewer 4: we are grateful for the constructive comments you provided, which helped us to improve the manuscript significantly.

Our responses to your comments are outlined below and highlighted in gray (to differentiate them from responses to other reviewers) in the new version.

Responses to Reviewer 4

  1. The authors must be added a section concerning the future perception of peri-implantitis treatment and antibiotic resistance.

Response:  The recommendation was included in the introduction. Part of the comments are shared with another reviewer; therefore, it is also highlighted in yellow.

  1. The authors must be added a section concerning the recommendation for peri-implantitis treatment.

Response: The recommendation was included in the introduction. Part of the comments are shared with another reviewer; therefore, it is also highlighted in yellow.

Round 2

Reviewer 1 Report

Accept

Author Response

Dear Reviewer 1:

We appreciate the acceptance of the manuscript for publication.

Reviewer 3 Report

You add 2 more databases and you didnt increase the number of studies identified. Despite you have duplicate articles the number of articles identified in two databases should be addresed. This is so serious and important. 

Author Response

Dear Reviewer 3: we are grateful for the constructive comments you provided.

Our response to your comment is outlined below and highlighted in yellow in the new version.

Response to Reviewer 3

  1. You add 2 more databases, and you did not increase the number of studies identified. Despite duplicate articles, the number of articles identified in the two databases should be addressed. This is so serious and important.

Response:  The recommendation was included. Corrections were made in the abstract, the first paragraph of the results, and the figure.
